# Submodular Maximization in Clean Linear Time

**Wenxin Li**
The Ohio State University
li.7328@osu.edu

**Moran Feldman**
University of Haifa
moranfe@cs.haifa.ac.il

**Ehsan Kazemi**
Google
ehsankazemi@google.com

**Amin Karbasi**
Yale University, Google Research
amin.karbasi@yale.edu.

## Abstract

In this paper, we provide the first deterministic algorithm that achieves $1/2$-approximation for monotone submodular maximization subject to a knapsack constraint, while making a number of queries that scales only linearly with the size of the ground set $n$. Moreover, our result automatically paves the way for developing a linear-time deterministic algorithm that achieves the tight $1 - 1/e$ approximation guarantee for monotone submodular maximization under a cardinality (size) constraint. To complement our positive results, we also show strong information-theoretic lower bounds. More specifically, we show that when the maximum cardinality allowed for a solution is constant, no deterministic or randomized algorithm making a sub-linear number of function evaluations can guarantee any constant approximation ratio. Furthermore, when the constraint allows the selection of a constant fraction of the ground set, we show that any algorithm making fewer than $\Omega(n/\log(n))$ function evaluations cannot perform better than an algorithm that simply outputs a uniformly random subset of the ground set of the right size. We extend our results to the general case of maximizing a monotone submodular function subject to the intersection of a $p$-set system and multiple knapsack constraints. Finally, we evaluate the performance of our algorithms on multiple real-life applications, including movie recommendation, location summarization, Twitter text summarization, and video summarization.

## 1 Introduction

In the era of big data, one of the main challenges that we daily face is to make an informed decision while taking into account all the relevant information. The study of how to make such near-optimal decisions from a massive pool of possibilities is at the heart of combinatorial optimization. It is well-known that without further structures, finding a near-optimal solution for a combinatorial optimization problem is notoriously hard, similar to finding a global minimum of a general non-convex function. Nevertheless, the problems of practical interest, such as the ones we encounter in machine learning applications, are often much more well-behaved which makes them amenable to (exact or approximate) optimization techniques.

One celebrated combinatorial structure is submodularity, also known as diminishing returns, stating that the added value of an element (e.g., image, sensor, etc.) to a context (data set of images, sensor network, etc.) decreases as the context in which it is considered grows. Submodular functions have attracted significant interest from the machine learning community, and have led to the development of algorithms with near-optimal solutions for a wide range of applications, including outbreak detection [52], graph cuts in computer vision [40], image and video summarization [22, 32, 59, 70], active learning [19, 27, 28], compressed sensing and structured sparsity [3, 16], fairness in machine

36th Conference on Neural Information Processing Systems (NeurIPS 2022).

learning [13, 41], recommendation [60], human brain parcellation [64], model training [57, 58], and learning causal structures [66, 75]. For recent surveys on the applications of submodular functions in machine learning and artificial intelligence, we refer the interested reader to [7, 69].

In their pioneering work, Nemhauser et al. [62] showed that a greedy algorithm achieves a tight $1 - 1/e$ approximation for *maximizing a monotone submodular function subject to cardinality constraint* (SMC). Running the greedy algorithm is too expensive as it requires $O(nk)$ function evaluations, where $n$ is the size of the ground set and $k$ is the maximum cardinality allowed by the constraint. Since then, there has been a significant effort to develop faster algorithms [12, 55, 72] and extend the approximation guarantee to more complex constraints (such as knapsack and $p$-set system) [4, 32].

In this work, we propose the first deterministic linear time algorithm for *maximizing a monotone submodular function subject to a knapsack constraints* (SMK), achieving $(1/2 - \varepsilon)$-approximation using $O_\varepsilon(n)$ function evaluations. We note that some nearly-linear time algorithms have been previously proposed for this problem. Ene and Nguyen [18] showed such an algorithm achieving $(1 - 1/e - \varepsilon)$-approximation, but the number of function evaluation used by their algorithm scales poorly with $\varepsilon$ (namely, $O((1/\varepsilon)^{1/\varepsilon})$), which makes it impractical even for moderate values of $\varepsilon$. This has motivated Yaroslavtsev et al. [73] to design an algorithm called `Greedy+` that achieve $(1/2 - \varepsilon)$-approximation and has a much more practical nearly-linear time complexity. Nevertheless, our algorithm manages to combine the approximation guarantee of `Greedy+` with a clean linear time. For the special case of a cardinality constraint, i.e., when the elements of the ground set have the same cost, our algorithm automatically reduces to a *deterministic* linear-time algorithm that guarantees $(1 - 1/e - \varepsilon)$ approximation via $O_\varepsilon(n)$ function evaluations. This special result has also been independently obtained by [46, 47]. Previously, the state-of-the-art *deterministic* algorithm only managed to achieve this approximation ratio using $O_\varepsilon(n \log \log n)$ function evaluations [36] (albeit in their new arXiv version, [38] claimed a linear-time algorithm, but with a worse dependence on $\varepsilon$ compared to our work).

Since the linear time complexity of our algorithm is the minimum required to read all the elements of the ground set, it is difficult to believe that any algorithm can improve over it while guaranteeing a reasonable approximation ratio. However, note that a single function evaluation can provide information about many elements of the ground set, and therefore, it is not a priori clear that an algorithm making a sub-linear number of function evaluations may not exist (even if its total time complexity is linear). Nevertheless, we are able to prove that this is indeed the case in at least two important regimes. First, we show that when the maximum cardinality allowed for the solution is constant, no algorithm making a sub-linear number of function evaluations can guarantee any constant approximation ratio for SMC. Second, when the constraint allows the selection of a constant fraction of the ground set, we show that any algorithm making fewer than $\Omega(n/\log(n))$ function evaluations cannot perform better than an algorithm that simply outputs a uniformly random subset of the ground set of the right size. Our technique also leads to a similar inapproximability result for *unconstrained submodular maximization* (USM). These lower bounds, along with the corresponding linear-time algorithms, nearly complete our understanding of the minimum query complexity of SMC, SMK, and USM problems. It is important to note that our inapproximability results are based on information theoretic arguments rather than relying on any complexity assumptions.

We consider the more general setting where we *maximize a monotone submodular function subject to a $p$-set system and $d$ knapsack constraints* (SMKS). We study the trade-off between the time complexity and approximation ratio for this problem. In particular, our results improve over the state-of-the-art approximation for nearly-linear time algorithms [4]. Finally, we evaluate the performance of our algorithms using real-life and synthetic experiments, including movie recommendation, location summarization, vertex cover, twitter text summarization and video summarization.

## 1.1 Additional Related Work

The work of Badanidiyuru and Vondrák [4] is probably the first work whose main motivation was to find algorithms with guaranteed approximation ratios for submodular maximization problems that provably enjoy faster time complexities. This work introduced the thresholding technique, which became a central component of basically every fast deterministic implementation of the greedy algorithm considered to date, including our algorithms. In particular, using this technique, Badanidiyuru and Vondrák [4] described an $O_\varepsilon(n \log n)$ time algorithm for maximizing a monotone submodular function subject to a cardinality constraint.

Sviridenko [67] was the first to obtain $(1 - 1/e)$-approximation for maximizing a monotone submodular function subject to a knapsack constraint. Their original algorithm was very slow (having a time complexity of $O(n^5)$), but it can be improved to run in $\tilde{O}(n^4)$ time using the thresholding technique of [4]. The algorithm of Sviridenko [67] is based on making 3 guesses. Recently, Feldman et al. [24] showed that the optimal approximation ratio of $1 - 1/e$ can be obtained using only 2 guesses, leading to $\tilde{O}(n^3)$ time. They also showed that a single guess suffices to guarantee a slightly worse approximation ratio of $0.6174$ and requires only $\tilde{O}(n^2)$ time [24]. There exists also an impractical nearly-linear time algorithm for the problem due to [18]. Another fast algorithm was suggested by [4], but an error was later found in their analysis (see [18] for details). We also note that multiple recent works aimed to bound the performance of a natural (and historically important) greedy algorithm [24, 49, 68]. A brief review on non-monotone submodular maximization subject to cardinality and knapsack constraints can be found in Appendix A.

When considering fast algorithms, it is important to consider also (semi-)streaming algorithms since these algorithms are usually naturally fast. Among the first semi-streaming algorithms for submodular maximization was the work of Badanidiyuru et al. [5], who obtained $(1/2 - \varepsilon)$-approximation for SMC. While this work was later improved by Kazemi et al. [42] and extended to non-monotone functions [1], it was also proved that no semi-streaming algorithm can provide a better than $1/2$-approximation for this problem [23]. The more general case of a knapsack constraint was first considered in the semi-streaming setting by Yu et al. [74]. Improved algorithms for this problem were later found by [37, 39] for single pass and by [38, 73] for multiple passes.

Maximizing a monotone submodular function subject to a $p$-set system constraint was first studied by Fisher et al. [25], who showed that a natural greedy algorithm obtains $(p + 1)^{-1}$-approximation for this problem. Many years later, Badanidiyuru and Vondrák [4] used their thresholding technique to get a nearly-linear time algorithm guaranteeing $(p + 2d + 1 + \varepsilon)^{-1}$-approximation even when there are $d$ knapsack constraints (in addition to the $p$-set system constraint). Recently, Badanidiyuru et al. [6] presented a new technique that allows them to improve over the last approximation ratio. They explicitly consider the implications of their ideas only for the regime where $d \leq p$, and in this regime they achieve $[2(p + 1 + \varepsilon)]^{-1}$-approximation and $(p + 1 + \varepsilon)^{-1}$-approximation in $\tilde{O}(nr/\varepsilon)$ and $\tilde{O}(n^3 r^2/\varepsilon)$ time, respectively, where $r \leq n$ is the rank of $p$-set system. We note, however, that some of the ideas of [6] can also be applied to the regime $d > p$, leading to $(p + d + 1)^{-1}$-approximation in $\tilde{O}(n^3)$ time.[1] One can observe that the improvement in the approximation ratio obtained by the new results of [6] comes at the cost of an increased time complexity. Our results in Section 5 further study this trade-off between the time complexity and approximation ratio and improve over the state-of-the-art approximation for nearly-linear time algorithms.

## 2 Notations and Preliminaries

A **set function** $f: 2^{\mathcal{N}} \to \mathbb{R}$ over a ground set $\mathcal{N}$ is a function that assigns a numeric value to every subset of $\mathcal{N}$. Given such a function, an element $u \in \mathcal{N}$ and a set $S \subseteq \mathcal{N}$, we denote by $f(u \mid S) \triangleq f(S \cup \{u\}) - f(S)$ the marginal contribution of $u$ to $S$ with respect to $f$. The set function $f$ is submodular if $f(u \mid S) \geq f(u \mid T)$ for every two sets $S \subseteq T \subseteq \mathcal{N}$ and element $u \in \mathcal{N} \setminus T$, and it is monotone if $f(u \mid S) \geq 0$ for every element $u \in \mathcal{N}$ and set $S \subseteq \mathcal{N}$. All the set functions considered in this paper are also non-negative. Additionally, given two sets $S, T \in \mathcal{N}$, we define the shorthand $f(T \mid S) \triangleq f(S \cup T) - f(S)$. Furthermore, given a set $S$ and element $u$, we often use $S + u$ as a shorthand for $S \cup \{u\}$.

We assume throughout the paper that algorithms are able to access the objective function $f$ only through a value oracle that given a set $S \subseteq \mathcal{N}$ returns $f(S)$. This (standard) assumption allows us to prove inapproximability results that are based on information theoretic arguments, and therefore, are independent of any complexity assumptions.

A **set system** $\mathcal{M}$ is an ordered pair $(\mathcal{N}, \mathcal{I})$, where $\mathcal{N}$ is a ground set and $\mathcal{I}$ is a non-empty subset of $2^{\mathcal{N}}$ that obeys the following property: if $S \subseteq T \subseteq \mathcal{N}$ and $T \in \mathcal{I}$, then $S$ also belongs to $\mathcal{I}$. It is customary to refer to the sets of $\mathcal{I}$ as independent (and to sets that do not belong to $\mathcal{I}$ as dependent). A base of a set $S \subseteq \mathcal{N}$ is a subset of $S$ that is independent and is not included in any other set having

---

[1]To be more exact, to get the $(p + d + 1)^{-1}$-approximation one has to apply the idea used in the proof of Theorem 8 of [6] to the $(p + 2d + 1 + \varepsilon)^{-1}$-approximation algorithm due to [4].

these properties. The set system $\mathcal{M} = (\mathcal{N}, \mathcal{I})$ is called $p$-set system for an integer $p \geq 1$ if for every $S \subseteq \mathcal{N}$ the ratio between the sizes of the largest base of $S$ and the smallest base of $S$ is at most $p$.

## 3 Knapsack Constraint

In this section we consider the *Submodular Maximization subject to a Knapsack Constraint problem* (SMK). In this problem we are given a non-negative monotone submodular function $f: 2^{\mathcal{N}} \to \mathbb{R}_{\geq 0}$, a non-negative cost function $c: \mathcal{N} \to \mathbb{R}_{\geq 0}$ and a budget $B > 0$. We say that a set $S \subseteq \mathcal{N}$ is *feasible* if it obeys $c(S) \leq B$ (where $c(S) \triangleq \sum_{u \in S} c(u)$), and the objective of the problem is to find a set maximizing $f$ among all feasible sets $S \subseteq \mathcal{N}$. For simplicity, we assume below that $c(u) \in (0, B]$ for every element $u \in \mathcal{N}$ and that $B = 1$. These assumptions are without loss of generality because (i) every element of $\mathcal{N}$ whose cost exceeds $B$ can be simply removed, (ii) every element of $\mathcal{N}$ whose cost is $0$ can be safely assumed to be part of the optimal solution ($f$ is monotone), and (iii) one can scale the costs to make the budget $B$ equal to $1$.

When the knapsack constraint of SMK happens to be simply a cardinality constraint, we get a problem known as *Submodular Maximization subject to a Cardinality Constraint* (SMC). Formally, in this problem we have an integer parameter $1 \leq k \leq n$, and we are allowed to output any set whose size is at most $k$. One can observe that SMC is a special case of SMK in which $c(u) = 1/k$ for every element $u \in \mathcal{N}$.

The first step of our proposed solution to solve SMK and SMC problems is getting an estimate $\Gamma$ of the value of an optimal solution (up to a constant factor). We show how this can be done in Appendix B. Then, in Section 3.1 we present a variant of the threshold-greedy algorithm of Badanidiyuru and Vondrák [4] that, unlike the original algorithm of [4], uses the estimate $\Gamma$ to reduce the number of thresholds that need to be considered. In Section 3.2, by combining the threshold-greedy variant from Section 3.1 with a post-processing step, we show how to achieve the state-of-the-art algorithm for SMK. The theoretical guarantee of this algorithm is given by Theorem 3.1.

**Theorem 3.1.** *For every $\varepsilon > 0$, there exists a deterministic $(1/2 - \varepsilon)$-approximation algorithm for* `Submodular Maximization subject to a Knapsack Constraint (SMK)` *that uses* $O(n\varepsilon^{-1} \log \varepsilon^{-1})$ *time.*

Complementary to our result for SMK, our proposed variant of the threshold-greedy algorithm leads to a deterministic linear algorithm for SMC, which is given by the next theorem.

**Theorem 3.2.** *For every $\varepsilon > 0$, there exists a deterministic $(1 - 1/e - \varepsilon)$-approximation algorithm for* `Submodular Maximization subject to a Cardinality Constraint (SMC)` *using* $O(n/\varepsilon)$ *time.*

### 3.1 Fast Threshold Greedy Algorithm

In this section, we present Algorithm 1 as a variant of the threshold-greedy algorithm from [4]. Algorithm 1 uses $\Gamma$, our estimate of the optimal value ($\Gamma$ is calculated by Algorithm 4 which is given in Appendix B). Algorithm 4 enables us to reduce the number of thresholds that needed to be considered. Specifically, Algorithm 1 considers an exponentially decreasing series of thresholds between $8\alpha \cdot \Gamma$ and $\Gamma/e$, where $\alpha \geq 1$ is a parameter of the algorithm. For every threshold considered, the algorithm adds to its solution every element such that (i) adding the element to the solution does not violate feasibility, and (ii) the density of the element with respect to the current solution exceeds that threshold. In addition to the parameter $\alpha$, Algorithm 1 gets a quality control parameter $\varepsilon \in (0, 1)$.

The time complexity of Algorithm 1 is given by the next observation.

**Observation 3.3.** *The time complexity of Algorithm 1 is $O(n\varepsilon^{-1} \log \alpha)$.*

Due to space constraints, the proof of this observation and the other proofs omitted from this section can be found in Appendix C.

Let $\ell$ be the value of $h$ when Algorithm 1 terminated. We analyze the approximation ratio of Algorithm 1 by describing two lower bounds on the performance of this algorithm. The following lemma gives the simpler among these bounds. Its proof is based on the observation that the elements of $T \setminus S_\ell$ were not added by Algorithm 1 to its solution even during the outer loop's last iteration.

---
**Algorithm 1:** FAST THRESHOLD GREEDY($\varepsilon, \alpha$)
---
**1** Let $\Gamma$ (an estimate of the value of an optimal solution) be the output of Algorithm 4.
**2** Let $\tau \leftarrow 8\alpha \cdot \Gamma$, $h \leftarrow 0$ and $S_0 \leftarrow \varnothing$.
**3** **while** $\tau > (1 - \varepsilon)\Gamma/e$ **do**
**4**     **for** *every element* $u \in \mathcal{N}$ **do**
**5**        **if** $u \notin S_h$, $c(u) + c(S_h) \leq 1$ *and* $\frac{f(u|S_h)}{c(u)} \geq \tau$ **then**
**6**           Let $u_{h+1} \leftarrow u$ and $S_{h+1} \leftarrow S_h + u_{h+1}$.
**7**           Increase $h$ by 1.
**8**     Update $\tau \leftarrow (1 - \varepsilon)\tau$.
**9** **return** $S_h$.
---

**Lemma 3.4.** *For every set* $T \subseteq \mathcal{N}$, *if* $\max_{u \in T} c(u) \leq 1 - c(S_\ell)$, *then* $f(S_\ell) \geq f(T) - \frac{c(T)}{e} \cdot f(OPT)$. *In particular,* $\max_{u \in OPT} c(u) \leq 1 - c(S_\ell)$ *implies* $f(S_\ell) \geq (1 - 1/e) \cdot f(OPT)$.

The following lemma gives the other lower bound we need on the performance of Algorithm 1. The proof of this lemma is based on the standard argument in greedy-like algorithms, namely, the observation that adding all the elements of $T$ to a solution increases the value of the solution to at least $T$, and therefore, at least one element of $T$ is a good choice that the algorithm can choose.

**Lemma 3.5.** *For every set* $\varnothing \neq T \subseteq \mathcal{N}$ *and integer* $0 \leq h < \ell$, *if* $\max_{u \in T} c(u) \leq 1 - c(S_h)$, *then* $\frac{f(S_{h+1}) - f(S_h)}{c(u_{h+1})} \geq \min\{(1 - \varepsilon) \cdot \frac{f(T|S_h)}{c(T)}, \alpha \cdot f(OPT)\}$.

Using the last two lemmata we can prove the following theorem, which implies Theorem 3.2. In a nutshell, the proof of this theorem uses Lemma 3.4 when the output of Algorithm 1 has the maximum allowed cardinality, and Lemma 3.5 otherwise.

**Theorem 3.6.** *If we choose* $\alpha = 1$, *then, for every* $\varepsilon \in (0, 1)$, *Algorithm 1 runs in* $O(n/\varepsilon)$ *time and guarantees* $(1 - 1/e - \varepsilon)$-*approximation for* SMC.

### 3.2 General Knapsack Constraint in Clean Linear Time

In this section we prove Theorem 3.1. We should point out that Algorithm 1 by itself does not guarantee any constant approximation guarantee for general knapsack constraints.[2] However, this can be fixed by using the post-processing step described by Algorithm 2. Recall that Algorithm 1 maintains a solution $S_k$ that grows as the algorithm progresses. The post-processing step takes $O(\varepsilon^{-1} \log \varepsilon^{-1})$ snapshots of this solution at various times during the execution of Algorithm 1, and then tries to augment each one of these snapshots with a single element.

---
**Algorithm 2:** FAST THRESHOLD GREEDY + POST-PROCESSING($\varepsilon$)
---
**1** Execute Algorithm 1 with $\alpha = \varepsilon^{-1}$. Store all sets $S_0, S_1, \ldots, S_\ell$ produced by this execution.
**2** **for** $i = 0$ **to** $\lfloor \log_{1+\varepsilon} \varepsilon^{-1} \rfloor$ **do**
**3**     Let $S^{(i)}$ be the set $S_k$ for the maximal $k$ value for which $c(S_k) \leq \varepsilon(1 + \varepsilon)^i$.
**4**     **if** *there exists an element* $u \in \mathcal{N}$ *such that* $c(u) \leq 1 - c(S^{(i)})$ **then**
**5**        Let $u^{(i)}$ be an element maximizing $f(u^{(i)} \mid S^{(i)})$ among all the elements obeying the condition on the previous line.
**6**        Let $S^{(i)+} \leftarrow S^{(i)} + u^{(i)}$.
**7**     **else** Let $S^{(i)+} \leftarrow S^{(i)}$.
**8** **return** *the set maximizing* $f$ *in* $\{S_\ell\} \cup \{\{u\} \mid u \in \mathcal{N}\} \cup \{S^{(i)+} \mid 0 \leq i \leq \lfloor \log_{1+\varepsilon} \varepsilon^{-1} \rfloor\}$.
---

The time complexity of Algorithm 2 is given by the next observation.

---

[2]To see this, consider an input instance consisting of an element $u$ of size $\delta > 0$, an additional element $w$ of size 1 and a function $f(S) = 2\delta \cdot |S \cap \{u\}| + |S \cap \{w\}|$. For small enough $\varepsilon$ and $\delta$ values, Algorithm 1 will pick the solution $\{u\}$ of value $2\delta$ instead of the much more valuable solution $\{w\}$ whose value is 1.

**Observation 3.7.** *Algorithm 2 runs in $O(n\varepsilon^{-1}\log\varepsilon^{-1})$ time.*

The rest of this section is devoted to analyzing the approximation ratio of Algorithm 2. Consider the inequality

$$f(S^{(i)+}) < \tfrac{1}{2}\cdot f(OPT) \qquad \forall\, 0 \le i \le \log_{1+\varepsilon}\varepsilon^{-1} \ . \tag{1}$$

If this inequality does not hold for some integer $i$, then Algorithm 2 clearly obtains at least $\tfrac{1}{2}$-approximation. Therefore, it remains to consider the case in which Inequality (1) applies for every integer $0 \le i \le \log_{1+\varepsilon}\varepsilon^{-1}$. One can observe that this inequality implies that $OPT$ is non-empty because $f(S^{(0)}) \ge f(\varnothing)$ by the monotonicity of $f$, and therefore, we can define $r$ to be an element of $OPT$ maximizing $c(r)$. Additionally, we assume below that $r$ is not the only element of $OPT$—if this assumption does not hold, then Algorithm 2 trivially returns an optimal solution.

**Lemma 3.8.** *If $c(r) \ge 1-\varepsilon$, then $f(S_\ell) \ge (\tfrac{1}{2}-\varepsilon)\cdot f(OPT)$.*

If $f(\{r\}) \ge \tfrac{1}{2}\cdot f(OPT)$, then the last lemma follows immediately since $\{r\}$ is one of the sets considered for the output of Algorithm 2 on Line 8. Otherwise, we get that $OPT \setminus \{r\}$ is a set with a lot of value taking a very small part of the budget allowed, and the proof of Lemma 3.8 uses Lemmata 3.4 and 3.5 to argue that $f(S_\ell)$ is always large when such a set exists.

So far we have proved that the approximation ratio of Algorithm 2 is at least $\tfrac{1}{2}-\varepsilon$ when either Inequality (1) is violated or $c(r) \ge 1-\varepsilon$. The following lemma proves that the same approximation guarantee applies also when neither of these cases applies, which completes the proof of Theorem 3.1. The proof of this lemma defines $i_r$ to be the maximal integer such that $\varepsilon(1+\varepsilon)^{i_r} \le 1-c(r)$, and $h_r$ to be the index for which $S_{h_r} = S^{(i_r)}$. If $h_r = \ell$, then the proof uses Lemma 3.4 to argue that $f(S_{h_r})$ is large. Otherwise, the proof uses Lemma 3.5 to argue that $f(S_{h_r+1})$ is large.

**Lemma 3.9.** *If $c(r) \le 1-\varepsilon$, then Inequality (1) implies $f(S_\ell) \ge (\tfrac{1}{2}-\varepsilon)\cdot f(OPT)$.*

## 4 Information-Theoretic Inapproximability Results

In this section we present our inapproximability results. The first result that we show is inapproximability for SMC (see Section 3 for the definition of this problem) showing that, when the number $k$ of elements allowed in the solution is constant, one cannot obtain a constant approximation ratio for this problem using $o(n)$ value oracle queries.

**Theorem 4.1.** *Any (possibly randomized) algorithm guaranteeing $\alpha$-approximation ($\alpha \in (0,1]$) for* Submodular Maximization subject to a Cardinality Constraint (SMC) *must use $\Omega(\alpha n/k)$ value oracle queries. In particular, this implies that the algorithm must make $\Omega(n)$ value oracle queries when $\alpha$ and $k$ are constants.*

See Appendix D for the proof of Theorem 4.1. Let us now consider the approximability of SMC in a different regime allowing larger values of $k$. Specifically, we consider the regime in which the ratio between $n$ and $k$ is some constant $\beta$. In this regime one can obtain $\beta$-approximation by simply outputting a uniformly random subset of $\mathcal{N}$ of size $k$ (which requires no value oracle queries); thus, this regime cannot admit an extremely pessimistic inapproximability result like the one given by Theorem 4.1. Nevertheless, the following theorem shows that improving over the above-mentioned "easy" approximation guarantee requires a nearly-linear query complexity.

**Theorem 4.2.** *For any rational constant $\beta \in (0,1)$ and constant $\varepsilon > 0$, every (possibly randomized) algorithm guaranteeing approximation ratio of $\beta + \varepsilon$ for instances of* Submodular Maximization subject to a Cardinality Constraint *obeying $k = \beta n$ must use $\Omega(\frac{n}{\log n})$ value oracle queries. Moreover, this is true even when the function $f$ of is guaranteed to be a linear function.*

We prove in Section E.1 a restricted version of Theorem 4.2 that applies only to deterministic algorithms. Then, Section E.2 extends the proof to randomized algorithms, which implies Theorem 4.2. Table 1 summarizes our knowledge about the number of value oracle queries necessary for obtaining various approximation guarantees for SMC.

Using the technique of Theorem 4.2, we can also prove another interesting result. In the *Unconstrained Submodular Maximization problem* (USM), the input consists of a non-negative (not necessarily monotone) submodular function $f\colon 2^{\mathcal{N}} \to \mathbb{R}_{\ge 0}$. The objective is to output a set $S \subseteq \mathcal{N}$ maximizing $f$. Our result for USM is given by the next theorem (proved in Appendix F). This theorem

Table 1: Number of value oracle queries that are required and sufficient to obtain any given approximation ratio for `Submodular Maximization subject to a Cardinality Constraint` (SMC). The variable $\varepsilon$ should be understood as an arbitrarily small constant.

| Regime | Approximation Ratio Range | Query Complexity |
|---|---|---|
| Constant $k$ | $[\varepsilon, 1 - 1/e - \varepsilon]$ | Can be obtained using $O(n)$ queries by Theorem 3.2 or Mirzasoleiman et al. [55], and requires $\Omega(n)$ queries by Theorem 4.1. |
| | $[1 - 1/e, 1]$ | Exact solution can be obtained using $O(n^k)$ queries via brute force; $\Omega(n)$ queries are necessary by Theorem 4.1. |
| $k = \beta n$ | $[0, \beta]$ | Can be obtained by outputting a uniformly random subset of the right size (0 queries). |
| | $[\beta + \varepsilon, 1 - 1/e - \varepsilon]$ | Can be obtained using $O(n)$ queries by Theorem 3.2 or Mirzasoleiman et al. [55]; requires $\Omega(\frac{n}{\log n})$ queries by Theorem 4.2. |
| | $1 - 1/e$ | Obtained by Greedy using $O(nk)$ queries [62]; requires $\Omega(\frac{n}{\log n})$ queries by Theorem 4.2. |
| | $[1 - 1/e + \varepsilon, 1]$ | Requires exponentially many queries [61]. |

Table 2: Number of value oracle queries required and sufficient to obtain any given approximation ratio for `Unconstrained Submodular Maximization` (USM). The variable $\varepsilon$ should be understood as an arbitrarily small constant.

| Approximation Ratio Range | Required Query Complexity | Sufficient Query Complexity |
|---|---|---|
| $[0, 1/4]$ | 0 | 0 
 An approximation ratio of $1/4$ can be obtained by outputting a uniformly random subset of $\mathcal{N}$ [20]. |
| $[1/4 + \varepsilon, 1/2]$ | $\Omega(\frac{n}{\log n})$    (Theorem 4.3) | $O(n)$ 
 Using the Double Greedy algorithm of Buchbinder et al. [11]. |
| $[1/2 + \varepsilon, 1]$ | Requires an exponential number of queries [20]. | – |

nearly completes our understanding of the minimum query complexity necessary to obtain various approximation ratios for `USM`. See Table 2 for more detail.

**Theorem 4.3.** *For any constant $\varepsilon > 0$, any algorithm for `Unconstrained Submodular Maximization` that guarantees an approximation ratio of $1/4 + \varepsilon$ must use $\Omega(\frac{n}{\log n})$ value oracle queries.*

The necessity of the $\log n$ term in Theorems 4.2 and 4.3 is discussed in Appendix G. Specifically, we show in this appendix an algorithm that can be used to solve (exactly) the hard problem underlying these theorems using $O(n/\log n)$ oracle queries. This algorithm is not efficient in terms of its time complexity, but it shows that one cannot prove a lower bound stronger than $\Omega(n/\log n)$ for this problem based on information theoretic arguments alone.

## 5   Set System and Multiple Knapsacks Constraints

In this section we consider the *Knapsacks and a Set System Constraint problem* (SMKS). In this problem we are given a non-negative monotone submodular function $f: 2^{\mathcal{N}} \to \mathbb{R}_{\geq 0}$, a $p$-system $\mathcal{M} = (\mathcal{N}, \mathcal{I})$, $d \geq 1$ non-negative cost functions $c_i: \mathcal{N} \to \mathbb{R}_{\geq 0}$ (one function for every integer $1 \leq i \leq d$) and $d$ positive values $B_1, B_2, \ldots, B_d$ as Knapsack budgets. We say that a set $S \subseteq \mathcal{N}$ is *feasible* if it is independent in $\mathcal{M}$ and also obeys $c_i(S) \leq B_i$ for every $1 \leq i \leq d$ (where $c_i(S) \triangleq \sum_{u \in S} c_i(u)$). The objective of the problem is to find a set maximizing $f$ among all feasible sets $S \subseteq \mathcal{N}$.

Algorithm 3 is the basic version of our proposed solution for `SMKS` (in Appendix I.1 we explain how to make this basic algorithm nearly-linear). Algorithm 3 gets two parameters. The first of

these parameters is an integer $\lambda \geq 1$. Elements that have a value larger than $\lambda^{-1}$ with respect to at least one function $c_i$ are considered big elements, and are stored in the set $B$ of the algorithm. The other elements of $\mathcal{N}$ are considered small elements. The algorithm never considers any solution that includes both big and small elements. Instead, it creates one candidate solution $S_B$ from the big elements, and one candidate solution from the small elements, and then outputs the better among the two (technically, in some cases the algorithm outputs directly the candidate solution based on the small elements without comparing it to $S_B$). The candidate solution $S_B$ is constructed using a procedure called `BigElementsAlg` that gets the set $B$ as input and outputs a feasible set whose value is at least $\alpha \cdot f(OPT \cap B)$, where $\alpha$ is a some value in $(0, 1]$ and $OPT$ is an arbitrary optimal solution. At this point we ignore the implementation of `BigElementsAlg`, and leave the value of $\alpha$ unspecified. These gaps are filled in Section H.3.

Most of Algorithm 3 is devoted to constructing the candidate solution out of small elements, which we refer to as the "small elements solution". In the construction of this solution, Algorithm 3 uses its second parameter $\rho \geq 0$ that intuitively should represent the density of the small elements of $OPT$. The algorithm initializes the small elements solution to be empty, and then iteratively adds to it the element with the largest marginal contribution among the small elements that have two properties: (i) their addition to the solution does not make it dependent in $\mathcal{M}$, and (ii) their density (the ratio between their marginal contribution and cost according to the linear constraints) is at least $\rho$. This process of growing the small elements solution can end in two ways. One option is that the process ends because no additional elements can be added to the solution (in other words, no element has the two properties stated above). In this case the better among $S_B$ and the small elements solution obtained $S_k$ is returned. The other way in which the process of growing the small elements solution can end is when it starts violating at least one linear constraint. When this happens, Algorithm 3 uses a procedure called `SetExtract` (see Algorithm 7 in Section H.1) to get a subset of the small elements solution that is feasible and also has a good value, and this subset is returned.

Algorithm 3 assumes access to an estimate $\rho$ of the density of the small elements of an optimal solution for the problem. In Section H.2 we explain how the dependence of the algorithm on $\rho$ can be dropped without increasing the time complexity of the algorithm by too much. Recall that Section 1.1 demonstrated a trade-off between the time complexity and approximation guarantee of state-of-the-art algorithms for SMKS. The next theorem further studies this trade-off, and in particular, improves over the state-of-the-art approximation for nearly-linear time algorithms. The $\tilde{O}$ notation in this theorem suppresses factors that are poly-logarithmic in $n$, $d$ and $\varepsilon^{-1}$. Due to space constraints, the proof of Theorem 5.1 is differed to Appendix H.

**Theorem 5.1.** *For every $\varepsilon > 0$, there exist algorithms that achieve $[(1 + O(\varepsilon))(p + 1 + \frac{7}{4}d)]^{-1}$-approximation and $(p + 1.5556 + \frac{13}{9}d + \varepsilon)^{-1}$ for* `Submodular Maximization subject to Knapsacks and a Set System Constraint` *in $\tilde{O}(nd + n/\varepsilon)$ and $\tilde{O}(n^2 + nd)$ time, respectively.*

---

**Algorithm 3:** `Basic Algorithm`$(\lambda, \rho)$

---

// Build the set of big elements, and find a solution based on them.

**1** Let $B \leftarrow \{u \in \mathcal{N} \mid \exists_{1 \leq i \leq d} \, c_i(u) > \lambda^{-1}\}$.

**2** Let $S_B$ be the output set of `BigElementsAlg`$(B)$.

// Construct a solution from the small elements.

**3** Let $S_0 \leftarrow \varnothing, k \leftarrow 0$.

**4** **while** *there exists an element $u \in \mathcal{N} \setminus (S_k \cup B)$ such that $S_k + u \in \mathcal{I}$ and*
$f(u \mid S_k) \geq \rho \cdot \sum_{i=1}^{d} c_i(u)$ **do**

**5**     Let $v_{k+1}$ be an element maximizing $f(u \mid S_k)$ among all the elements obeying the condition of the loop.

**6**     Let $S_{k+1} \leftarrow S_k + v_{k+1}$.

**7**     **if** $\max_{1 \leq i \leq d} c_i(S_{k+1}) \leq 1$ **then** Increase $k$ by 1.

**8**     **else return** *the output set of* `SetExtract`$(\lambda, S_{k+1})$.

**9** **return** *the better set among $S_B$ and $S_k$.*

---

# 6 Experimental Results

We have studied the submodular maximization problem subject to different constraints. In this section, we compare our proposed algorithms with the state-of-the-art algorithms under the following constraints: (i) a carnality constraint (Section 6.1), (ii) a single knapsack constraint (Section 6.2), and (iii) combination of a $p$-system and $d$ knapsack constraints (Appendix J.4).

## 6.1 Cardinally Constraint

Cardinality constraint is the most widely studied setup in the submodular maximization literature. We compare our algorithm, FASTTHRESHOLDGREEDY (abbreviated as FTG) with LAZYGREEDY and BOOSTRATIO [46] under cardinality constraint. LAZYGREEDY is an efficient implementation of the naïve GREEDY algorithm in which the diminishing returns property of submodular functions is used to avoid oracle queries that are known to provide little gain [55]. It is well-known that LAZYGREEDY leads to several orders of magnitude speedups over GREEDY in practice.

In our first experiment, we implement a movie recommender system by finding a diverse set of movies for a user. We adopt the approach of Lindgren et al. [54] to extract features for each movie by using ratings from the MovieLens dataset [31]. For a given set of movies $\mathcal{N}$ ($|\mathcal{N}| = n$), let vector $v_i$ represents the attributes of the $i$-th movie. The similarity matrix $M_{n \times n}$ between movies is defined by $M_{ij} = e^{-\lambda \cdot \text{dist}(v_i, v_j)}$, where $\text{dist}(v_i, v_j)$ is the euclidean distance between vectors $v_i, v_j \in \mathcal{N}$. For this application, we used the following monotone and submodular function to quantify the diversity of a given set of movies $S$: $f(S) = \log \det(\mathbf{I} + \alpha M_S)$, where $\mathbf{I}$ is the identity matrix and $M_S$ is a square sub-matrix of $M$ consisting of the rows and columns corresponding to the movies in the set $S$. Our objective is to maximize $f$ under a cardinality constraint $k$. In Figure 1a, we compare the utility of the algorithms on this instance. We observe that FASTTHRESHOLDGREEDY with $\varepsilon \in \{0.1, 0.2\}$ performs as good as LAZYGREEDY, and BOOSTRATIO performs slightly worse. In Figure 1b, we observe that the query complexity of FASTTHRESHOLDGREEDY is significantly less than that of LAZYGREEDY. It is interesting to note that, as is guaranteed by our theoretical results, the number of oracle calls for FASTTHRESHOLDGREEDY is (almost) not increasing with $k$. In our second experiment, which appears in Appendix J.1, we consider a location summarization application and show that similar conclusions apply also to this application.

STOCHASTICGREEDY is a fast but randomized approach for solving SMC [55]. In Appendix J.2 we present an experiment comparing FASTTHRESHOLDGREEDY with STOCHASTICGREEDY under the cardinality constraint in a vertex cover context. In general, this experiment shows that the utility of FASTTHRESHOLDGREEDY is significantly better than that of STOCHASTICGREEDY, and that FASTTHRESHOLDGREEDY sometimes also outperforms STOCHASTICGREEDY in terms of the query complexity. We also observe a high variability in the utility of solutions returned by STOCHASTICGREEDY, which diminishes the benefit from this algorithm.

## 6.2 Single Knapsack Constraint

In this section, we evaluate the performance of Algorithm 2 (referred to as FTGP) with that of DYNAMICMRT [37] and DENSITYGREEDY under a single knapsack constraint. DENSITYGREEDY greedily adds elements that maximize the ratio between their marginal gain and knapsack cost (but ignoring elements whose addition will result in a violation of the knapsack constraint). In our first experiment for this constraint, we again consider the movie recommendation application from Section 6.1. The cost of each movie is defined to be proportional to the absolute difference between the rating of that movie and 10 (the maximum rating in iMDB). In this application, the goal is to find a diverse set of movies while guaranteeing that the total rating of the picked movies is high enough [6]. From Figures 1c and 1d, we observe that Algorithm 2 significantly outperforms the other two algorithms with respect to both the utility and number of oracle calls metrics. For our second experiment under the knapsack constraint, which appears in Appendix J.3, we consider Twitter text summarization. In this experiment we again get that Algorithm 2 surpasses the baseline algorithms.

# 7 Conclusion and Societal Impact

An algorithm for maximizing a submodular function is conventionally measured in terms of two quantities: a) approximation guarantee, i.e., how well it performs with respect to the (exponential time) optimum algorithm, and b) query complexity, i.e., how many function evaluations it requires.

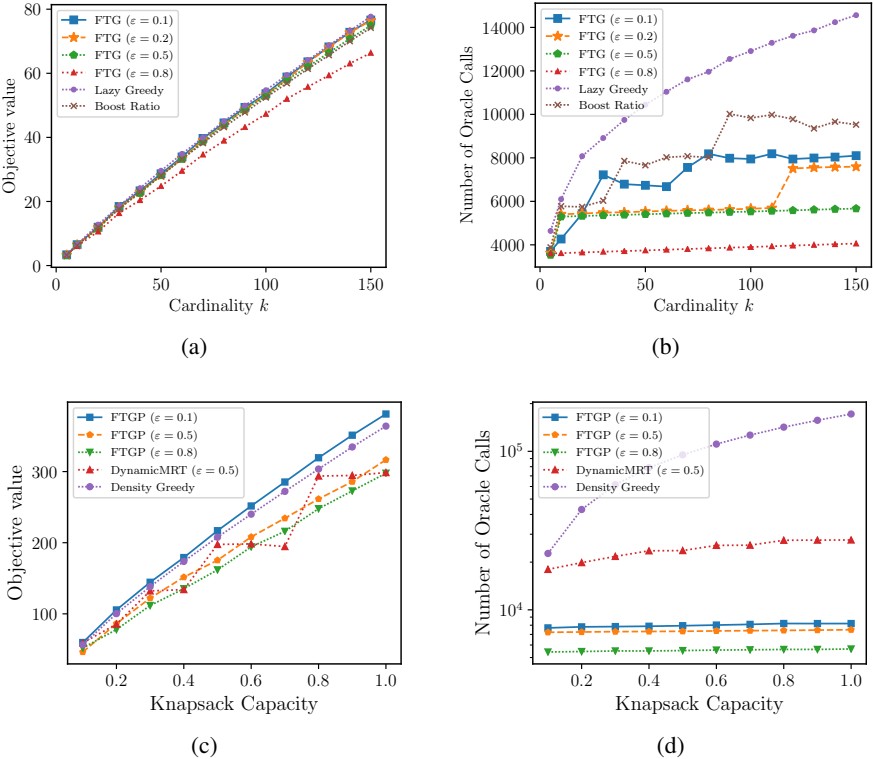

Figure 1: Movie Recommendation: (a) and (b) compare FASTTHRESHOLDGREEDY (referred to as FTG) with LAZYGREEDY and BOOSTRATIO [46] under a cardinality constraint; (c) and (d) compare Algorithm 2 (referred to as FTGP) with DYNAMICMRT [37] and DENSITYGREEDY under a single knapsack constraint.

In their seminal work, Nemhauser and Wolsey [61] and Nemhauser et al. [62] resolved part (a). In this paper, after 44 years and building on a large body of prior work, we nearly resolved part (b) and portrayed a nearly complete picture of the landscape. Specifically, we developed a clean linear-time algorithm for maximizing a monotone submodular function subject to a cardinality constraint (and more generally a knapsack constraint). We also provided information-theoretic lower bounds on the query complexity of both constrained and unconstrained (non-monotone) submodular maximization. Finally, we studied the trade-off between the time complexity and approximation ratio for maximizing a monotone submodular function subject to a $p$-system and $d$ knapsack constraints. This work does not have any negative societal impact.

Like most algorithms in the literature, our algorithms assume the ability to exactly evaluate the objective function. In some applications exact evaluations is too costly, which has motivated works (such as [33]) that considered erroneous value oracles. Extending our work in this direction is an interesting question for future research.

## Funding Transparency Statement

**Funding in direct support of this work:** The work of Moran Feldman was supported in part by Israel Science Foundation (ISF) grant no. 459/20. The work of Amin Karbasi was supported in part by the NSF (IIS-1845032), ONR (N00014- 19-1-2406), and the AI Institute for Learning-Enabled Optimization at Scale (TILOS).

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
