# OpenReview forum: "Submodular Maximization in Clean Linear Time"
_NeurIPS.cc/2022/Conference — NeurIPS 2022 Accept_

### Official Review · Reviewer_d5nd · 2022-07-03

**Rating:** 4
**Confidence:** 4
**Soundness:** 3 good
**Presentation:** 3 good
**Contribution:** 2 fair

**Summary:**

The paper studies the problem of submodular maximization under cardinality/knapsack constraints and provides fast $\mathit{deterministic}$ algorithm for the problem. The submodular maximization problem is a well-studied problem, where a set function $f$ defined over a ground set of n element, the set function is assumed to monotone and submodular (i.e., $f(A \cup \{u\} - f(A) \geq f(B \cup \{u\}) - f(B)$ for any set $A\subseteq B \subseteq [n]$ and element $\{u\}$). The goal is to maximize the function value, for cardinality constraints, the algorithm is allowed to select up to k elements, while for knapsack constraints, the algorithm is allowed to select ground set element with a knapsack constraints (each element is associated with some cost).

To summarize, the paper designs deterministic algorithm that achieves

(1) Under a knapsack constraint, it achieves $(1/2 -\epsilon)$ approximation in $O(n\epsilon^{-1}\log(1/\epsilon))$ times. The approximation ratio is not the state of art, as mentioned in the paper, as [Ene and Nguyen ICALP'19] proves a $(1-1/e -\epsilon)$  approximation algorithm, with linear time, but the dependency on $\epsilon$ is bad, i.e., $O((1/\epsilon)^{\epsilon})$.

(2) Under cardinality constraints, it achieves $(1-1/e-\epsilon)$-approximation using $O(n/\epsilon)$ oracles. The approximation ratio is optimal, but the stochastic greedy algorithm (a randomized algorithm) has better oracle complexity of $O(n\log(1/\epsilon))$. A nearly lower bound is also proved in the paper (but I am not sure whether these are folklores...)

(3) Constant approximation under multiple knapsack constraints.

(4) Extensive experiments showing the effectiveness of the algorithm

From a technical perspective, the algorithm is based on greedy algorithm with decreasing threshold. While for knapsack constraints, post processing is needed.



---------------------- Post rebuttal ------------------
After a few rounds of discussion with the author, I believe I convince the author that my main concern is (1) the significance of lower bound (I still believe it is more like a folklore result, but nice effort to write it down) and (2) the deterministic algorithm provides in the paper is not so exciting for an acceptance (I understand there are two or three paper in the area to develop deterministic algorithm).




**Questions:**

In line 365--366, it is claimed that the paper fully resolve the landscape of oracle complexity of submodular maximization, it seems too strong for me. After all, only deterministic algorithm is considered in this paper.

**Strengths And Weaknesses:**

Strength. The algorithm presents state of art $\mathit{deterministic}$ algorithm for the classic problem of submodular maximization with cardinality constraints and a knapsack constraint. The experiments show the algorithm is practical, which is very nice!


Weakness. There is no reason one restricts to deterministic instead of randomize algorithm, this decreases the value of the paper a lot.

---

> ### Author Response · Authors · 2022-08-02
> **Why deterministic algorithms?**
>
>
> We would like to emphasize again that our lower bound results hold for both deterministic and randomized algorithms, while our algorithmic results are for the deterministic setting. Therefore, we are achieving the best of both worlds. The reason that we think deterministic algorithms are better than randomized one (aside from the fact that they do not have a probability of failure, which is nice but usually not very significant since this probability can be made very small) is that they always produce the same result (and there is in fact the entire field of pseudo-deterministic algorithms that tries to mimic that with randomized algorithms). We are interested in understanding the fundamental limits of deterministic algorithms, i.e, how many value oracle queries are necessary to achieve the near optimal approximation ratio. Given the existing obstacles for using derandomization techniques to submodular maximization problems, direct design of deterministic algorithms with desirable performance guarantee is of great interest to the community.
>
> Indeed in the existing literature, there are several works focusing on designing deterministic algorithms for submodular maximization problems. For example:
>
> [Ref 1] Niv Buchbinder , and Moran Feldman. Deterministic algorithms for submodular maximization problems. In SODA 2016.
>
> [Ref 2] Niv Buchbinder , Moran Feldman, and Mohit Garg. Deterministic $(1/2+ \varepsilon)$-Approximation for Submodular Maximization over a Matroid. In SODA 2019.
>
> [Ref 3] Alan Kuhnle. Interlaced greedy algorithm for maximization of submodular functions in nearly linear time. In NeurIPS 2019.
>
> [Ref 4] Kai Han, Shuang Cui, and Benwei Wu. Deterministic approximation for submodular maximization over a matroid in nearly linear time. In NeurIPS 2020.

---

> ### Author Response · Authors · 2022-08-02
> **Novelty of the lower bounds**
>
> We believe that our lower bound results for cardinality constraint are original and non-trivial. When $k$ is a constant fraction of $n$, approximation ratio no more than this constant requires $0$ queries, but any ratio larger than this constant requires nearly linear number of queries. This “threshold phenomena” should be new to the community. If you check the appendix, you will see how non-trivial the proofs are. If the reviewer believes that these results are folklores, we kindly ask for a reference.

---

> ### Author Response · Authors · 2022-08-02
> **Resolving the landscape of oracle complexity of submodular maximization**
>
> We would like to thank the reviewer for their feedback. We believe that the reviewer did not fully grasp the implications of our results and **incorrectly** believes that only deterministic algorithms are considered in the paper. Please see our responses below to your questions. We hope that they clarify our contributions.
>
> **On the lower bounds and oracle complexity**
>
> We should emphasis that our lower bounds for cardinality constraint and unconstrained maximization are information-theoretic, and they hold for both deterministic and randomized algorithms. More specifically, for a monotone submodular function subject to cardinality constraint we prove that, when the maximum cardinality allowed for the solution is constant, no algorithm (neither deterministic nor randomized) making a sub-linear number of function evaluations can guarantee any constant approximation ratio. Indeed we have discussed this aspect of the lower bounds in detail in the manuscript (see lines 60--75 of the submission). For more information on the nature of the lower bounds, and how we achieve them, please refer to Appendix D.2 and Appendix E. Therefore, we correctly claim that this paper fully resolve the landscape of oracle complexity of submodular maximization.

---

> ### Comment · Reviewer_d5nd · 2022-08-05
> **Reply to the author**
>
> Thanks for your reply.
>
> I want to clarify that I fully understand the lower bound result, i.e., it holds for randomize algorithm. I think it is a folklore result because I believe an expert in this field could figure out it very soon. I believe it is worth to write down and publish, but the claim of fully resolving the landscape is definitely too aggressive (and actually the lower and upper bound is not exactly matched, up to a factor of $k$). I strongly suggest the author to reword a little bit, it would not hurt the contribution of your paper.
>
> I also understand the effort of developing deterministic algorithm, but I don't think the current result is significant above the bar, given its worse approximation (or runtime) guarantee. I like  the fact it works well in practice though.

---

> > ### Author Response · Authors · 2022-08-07
> > **The tightness of results**
> >
> > We are happy that the reviewer agrees that our lower bounds hold for both deterministic and randomized algorithms. Therefore, no algorithm can achieve a better query complexity than the lower bounds we proved in our paper. We are also happy that the reviewer agrees that the results are worth publishing.
> >
> > Regarding the tightness of our results, and in the case of monotone submodular maximization subject to a cardinality constraint, please note that when k is constant, the lower and upper bounds match. When k is a linear fraction of n, then the lower and upper bounds match up to a log(k) factor (and **not** up to a factor of k as the reviewer suggested).  In fact, due to this logarithmic mismatch, we say in the conclusion that we portrayed a **nearly** complete picture of the landscape. As we are not aware of these folklore results the reviewer is referring to, we are unable to respond to such claims. Again, please let us know if there are any references that we are missing. Indeed, there are works in the literature trying to achieve submodular maximization in sub-linear time (see for example "Probabilistic Submodular Maximization in Sub-Linear Time"). Our work shows that such results, without further strong assumptions, cannot be achieved.
> >
> > Finally, it appears to us that the main complaint of the review is about the line in our conclusion *"we portrayed a nearly complete picture of the landscape."* We strongly feel this line is justified under the classical worst-case complexity point of view. However, there are indeed other notions of computational complexity such as average-case complexity (which is unrelated to the distinction between deterministic and randomized algorithms) that are very interesting directions for future work, and are beyond the scope of our paper. We can certainly include a remark along these lines in the conclusion if this is what the reviewer is suggesting.

---

> > > ### Comment · Reviewer_d5nd · 2022-08-08
> > > **Reply to the tightness of lower bound**
> > >
> > > Thanks for your reply.
> > >
> > > It is nothing to do with the sentence, it just my personal suggestion. Anyway, it is hard to believe that many people in the area (including a few well-known people in TCS area) does not know the lower bound. The only reason I can imagine is that they are aiming for a tighter characterization (resolve the dependence on $k$).

---

> > > > ### Comment · Area_Chair_6KyE · 2022-08-08
> > > > **References**
> > > >
> > > > Dear reviewer d5nd,
> > > >
> > > > You refer to the lower bound as being folklore and already known. If this is the case, could you provide a reference mentioning this?
> > > >
> > > > Thank you for your review and discussion about this paper.

---

### Official Review · Reviewer_FKqZ · 2022-07-05

**Rating:** 6
**Confidence:** 3
**Soundness:** 4 excellent
**Presentation:** 3 good
**Contribution:** 3 good

**Summary:**

In this submission, the problem of maximizing a submodular function subject to a knapsack constraint is considered, where the function is accessed by an oracle. There is a lot of literature on this problem and several approximation algorithms are known. In this paper, an approximation algorithm with a guarantee of 1/2-eps is developed whose running time is linear in n. This does not improve the best previously known approximation factor (which is 1-1/e) but it is the first algorithm with this approximation factor that runs in linear time. For the special case of a cardinality constraint the factor improves to 1-1/e-eps.

The algorithmic result is complemented by information-theoretic lower bounds. It is shown that a linear number of function evaluations are needed to obtain a constant approximation guarantee even in the case of a cardinality constraint. This lower bound is true if the number k of allowed elements is constant. If it is a constant fraction of the size n of the ground set then Omega(n/log(n)) function evaluations are shown to be necessary to achieve a constant approximation.

The theoretical results are supported by experimental results showing that the designed approximation algorithm outperforms baseline algorithms from the literature.

**Questions:**

l 40: "submodualr" -> "submodular"
l 50: "makes is impractical" -> "makes is impractical"
l 130: definition of f(T|S): I think f(T) should be f(S).
l 183: "where" -> "were"
page 15, last line: "forall e in ..." -> "forall u in"

**Limitations:**

n.a.

**Strengths And Weaknesses:**

The algorithm is very clean and natural and the proof of the approximation guarantee is easy to follow. I very much enjoyed reading this part of the submission. However, this part is also close to existing literature and many ideas in this part are taken from existing literature. A weakpoint is, of course, that the approximation ratio is only 1/2-eps and not 1-1/e-eps (which is known to be achievable in polynomial time). The lower bounds are rather involved but quite interesting. The experiments are clearly not the main focus of the submission and I would rather see them as a proof of concept that the algorithm works in practice but not as a thorough evaluation. For example, the two baselines are natural but it is never argued why exactly these algorithms are chosen as baselines and not other algorithms.

---

> ### Author Response · Authors · 2022-08-02
> **Five typos**
>
> Thanks for pointing these out, we have revised our paper accordingly.

---

> ### Author Response · Authors · 2022-08-02
> **How the baselines are chosen?**
>
> Our proposed algorithms for each constraint are compared with the state-of-the-art algorithms:
>
> * Cardinality constraint:
>     * LazyGreedy which is a quite efficient implementation of the naive greedy and BoostRatio which is the state-of-the-art for deterministic algorithms [36] (see Figure 1 (a,b) and Figure 2).
>     * StochasticGreedy which is a linear-time randomized algorithm (see Figure 3).
>
>     Both LazyGreedy and StochasticGreedy are used extensively in the previous works as fast and efficient baselines.
> * Knapsack  constraint:  DynamicMRT [28] (to the best of our knowledge it is the state-of-the-art algorithm for this problem) and DensityGreedy with lazy updates (see Figure 1 (c,d) and Figure 4). DensityGreedy is used in previous works such as [5, b] as efficient baselines. Note the the algorithm of [13] has an astronomical time complexity even for a moderate value of $\varepsilon$, and is thus, not appropriate as a baseline.
> * $p$-set system and $d$ knapsack constraints: BarrierGreedy [5] (to the best of our knowledge it is the state-of-the-art algorithm for this constraint), FAST [3], Greedy and DensityGreedy  (see Figure 5).
>
> It would be great if the reviewer could suggest better baselines for our comparisons. We would be more than happy to compare our algorithms with those suggestions in the revised version of the manuscript.
>
> **References:**
>
> [b] Baharan Mirzasoleiman, Ashwinkumar Badanidiyuru and Amin Karbasi. Fast constrained submodular maximization: Personalized data summarization. In ICML 2016.

---

> ### Author Response · Authors · 2022-08-02
> **Approximation guarantee of the algorithm for knapsack**
>
> The linear query complexity requirement leaves little room for using sophisticated approaches, which makes it hard to achieve the near optimal ratio of $1-1/e-\varepsilon$.

---

### Official Review · Reviewer_QGLr · 2022-07-06

**Rating:** 6
**Confidence:** 4
**Soundness:** 4 excellent
**Presentation:** 4 excellent
**Contribution:** 3 good

**Summary:**

The paper studies (mainly monotone) submodular maximization subject to various types of constraints. In particular, the authors focus is on deterministic algorithms that achieve constant factor approximation to the optimal solution while needing only linear time to terminate.

The main contributions of the paper are:
- a deterministic $(1-1/e-\varepsilon)$ approximation algorithm for monotone submodular maximization subject to cardinality constraint with $O(n/\varepsilon)$ running time
- a deterministic $(\frac 12 - \varepsilon)$ approximation algorithm for monotone submodular maximization subject to knapsack constraint with $O(n/\varepsilon \log(\frac{1}{\varepsilon}))$ running time

To complement these positive results, the authors construct information theoretical lower bounds that show how $\tilde\Omega(n)$ running time is needed to get constant factor approx for monotone submodular maximization with cardinality constraint and unconstrained non-monotone submodular maximization. For the former problem, it is also shown that when k is of the order of n, then no algorithm can do better than random sampling.

Finally, the authors extend their techniques and design fast algorithms for monotone submodular maximization with a very general class of constraints, namely set system and multiple knapsack constraints.


**Questions:**

In the experiments with knapsack, is density greedy implemented with lazy updates? If not, why? Using lazy updates brings the computational complexity from $O(n^2)$ to $O(n \log n)$

**Limitations:**

Other comments to the authors:
- please, state from the abstract that the positive results only apply to monotone objectives
- line 40 is submodular and not submodualr
- please mention in the preliminary that you are considering the value oracle model


**Strengths And Weaknesses:**

Achieving fast algorithms for submodular optimization is a rich and exciting line of research motivated by the applications and it is definitely within the scope of the conference.

As it is common in submodular maximization, many of the techniques used in this work are a careful combination of well known ideas and I do not think this is a limit of the paper. I particularly like Algorithm 4: although simple, it is crucial to argue that a constant number of thresholds is enough and I think it can be used in many other contexts. To the best of my knowledge it is original, but I might be mistaken. I also like the lower bounds: they are not surprising, but I am pretty sure they will be cited in future works.

On the negative side, the actual contribution of the paper seems a bit marginal. Linear time algorithms (although randomized) already exists for cardinality constraints, while for knapsack it is easy to see that a straightforward algorithm like density greedy + lazy updates already yields a constant factor approximation in O(n log n) iterations, so the improvement consists in shaving the extra log factor.

Although I have not checked in detail all the proofs, the claims are well supported. The paper is fairly well written and clear.

The related work section is very rich. I would just mention that the authors might want to spend some words on the case of non monotone objectives, for cardinality constraint see e.g. Kuhnle AAAI21, while for knapsack see Amanatidis et al. NeurIPS 20.

---

> ### Author Response · Authors · 2022-08-02
> **Contribution of the paper (regarding cardinality and knapsack constraints)**
>
> Firstly, we think deterministic algorithms are better than randomized one. The reason is that (aside from the fact that deterministic algorithms do not have a probability of failure, which is nice but usually not very significant since this probability can be made very small by running the algorithm several times) deterministic algorithms always produce the same result (and there is in fact the entire field of pseudo-deterministic algorithms that tries to mimic that with randomized algorithms). Furthermore, in Figure 3 (Appendix I.2), we have shown that our proposed deterministic algorithm for cardinality constraint performs better than the randomized StochasticGreedy algorithm in practice.
>
> Secondly, it is technically true that simply combining density greedy and lazy updates can obtain a constant factor approximation in $O(n \log n)$ queries, but the ratio will be significantly worse than $1/2$ (we believe one can get an approximation ratio of $(1 - 1/e)/2 - \varepsilon$ using this method via $O((n/\varepsilon) \log (n / \varepsilon))$ queries, so for any constant $\varepsilon$ this gives $O(n \log n)$ queries). More involved techniques are necessary to get better results than that. [Ref 1] gets $1/2-\varepsilon$ approximation using nearly-linear time, and the analysis is based on a new set of first-order linear differential inequalities and their robust approximate versions, so compared to them we really only shave a logarithmic factor. [Ref 2] is better than us from an asymptotic point of view, but has an astronomical time complexity even for a moderate value of $\varepsilon$. We reference the two papers and we are straightforward about both of these things in the introduction. It can be seen that these more involved techniques lead to a tradeoff between the approximation ratio and the practicality.  We are able to keep the approximation ratio and simplicity of the practical paper [Ref 1], while reducing the time complexity, which is likely to have a significant contribution in practice even if the theoretical improvement is only shaving the extra log factor.
>
> **References:**
>
> [Ref 1] Grigory Yaroslavtsev, Samson Zhou and Dmitrii Avdiukhin. "Bring Your Own Greedy"+Max: Near-Optimal 1/2-Approximations for Submodular Knapsack. In AISTAT 2020.
>
> [Ref 2] Alina Ene and Huy L. Nguyen. A Nearly-Linear Time Algorithm for Submodular Maximization with a Knapsack Constraint. In ICALP 2019.

---

> ### Author Response · Authors · 2022-08-02
> **Is density greedy implemented with lazy updates?**
>
> Yes, we have used the lazy update idea for the DensityGreedy algorithm. Furthermore, in order to be fair and provide a better comparison, in our experiments we also used a version of the BoostRatio algorithm [36] with lazy updates. Note that the original implantation provided for BoostRatio in [36] does not use lazy updates. In the attached code to the submission, we have provided the implementation of the algorithms with lazy updates (suffixed with *_optimized* in the code). In our experimental results, we have used those implementations.

---

> ### Author Response · Authors · 2022-08-02
> **Literature on non-monotone submodular objectives**
>
> Thanks for the suggestion, we will briefly review the literature related to non-monotone submodular maximization under cardinality and knapsack constraints. We will also make sure to include this complementary literature review in the revised version.
>
> **Non-monotone submodular maximization: literature review**
>
> For cardinality constraint, Buchbinder et al. [Ref 3] proposed the random greedy algorithm, which achieves an approximation ratio of $1/e$ using $O(nk)$ queries. Combining random greedy with a continuous double greedy algorithm, the ratio can be improved to $1/e+0.004$. By derandomizing the random greedy algorithm, Buchbinder and Feldman [Ref 4] shows that there exist an $1/e$-approximate deterministic algorithm. To improve the computational speed, an $(1/e-\varepsilon)$-approximate random sampling algorithm that makes $O(\frac{n}{\varepsilon^{2}}\log \frac{1}{\varepsilon})$ was proposed in [Ref 3]. Sakaue [Ref 6] shows that with slight modification, stochastic greedy can achieve almost 1/4-approximation in expectation in linear time. Based on the idea of interlacing two greedy procedures, Kuhnle [Ref 5] developed an $(1/4-\varepsilon)$-approximate deterministic algorithm, which requires $O(\frac{n}{\varepsilon} \log \frac{k}{\varepsilon})$ queries. Algorithms with low adaptive complexity were
> studied in [Ref 10]. Note that algorithms for the more general matroid constraint can be also applied to cardinality constraint.
>
> For knapsack constraint, there is a line of work using multi-linear extension of submodular function to obtain improved performance guarantees. Mirzasoleiman
> et al. [Ref 7] proposed a deterministic $(1/10-\varepsilon)$-approximate algorithm using $O(\frac{n^{2}\log n}{\varepsilon})$ queries. Currently the best deterministic algorithm is due to Han et al. [Ref 8], where an approximation ratio of $(1/6-\varepsilon)$ is achieved by $O(\frac{n}{\varepsilon}\log \frac{k}{\varepsilon})$ queries. There is also a randomized $1/(3+2\sqrt{2})$-approximate algorithm propose by Amanatidis et al. [Ref 9], which requires $O(\frac{n}{\varepsilon}\log \frac{n}{\varepsilon})$ queries. This was later improved by Han et al. [Ref 8], where an approximation ratio of $1/4$ is shown to be obtainable within $O(\frac{n}{\varepsilon}\log \frac{k}{\varepsilon})$ queries.
>
>
> **References:**
>
> [Ref 3] Niv Buchbinder, Moran Feldman, Joseph Naor and Roy Schwartz. Submodular maximization with cardinality constraints. In SODA 2014.
>
> [Ref 4] Niv Buchbinder and Moran Feldman. Deterministic Algorithms for Submodular Maximization Problems. In SODA 2016.
>
> [Ref 5] Alan Kuhnle. Interlaced Greedy Algorithm for Maximization of Submodular Functions in Nearly Linear Time. In NeurIPS 2019.
>
> [Ref 6]	Shinsaku Sakaue. Guarantees of Stochastic Greedy Algorithms for Non-monotone Submodular Maximization with Cardinality Constraint. In AISTAT 2020.
>
> [Ref 7] Baharan Mirzasoleiman, Ashwinkumar Badanidiyuru and Amin Karbasi. Fast Constrained Submodular Maximization: Personalized Data Summarization. In ICML 2016.
>
> [Ref 8] Kai Han, Shuang Cui, Tianshuai Zhu, Enpei Zhang, Benwei Wu, Zhizhuo Yin, Tong Xu, Shaojie Tang and He Huang. Approximation Algorithms for Submodular Data Summarization with a Knapsack Constraint. In Sigmetrics 2021.
>
> [Ref 9] Georgios Amanatidis, Federico Fusco, Philip Lazos, Stefano Leonardi, and Rebecca Reiffenh{\"a}user. Fast Adaptive Non-Monotone Submodular Maximization Subject to a Knapsack Constraint. In NeurIPS 2020.
>
> [Ref 10] Alan Kuhnle. Nearly Linear-Time, Parallelizable Algorithms for Non-Monotone Submodular Maximization. In AAAI 2021.

---

> ### Author Response · Authors · 2022-08-02
> **Clarify that the positive results apply to monotone objectives**
>
> Thanks for the suggestions. We will revise the abstract and the paper (fix the typo) accordingly.

---

### Official Review · Reviewer_Z74k · 2022-07-11

**Rating:** 8
**Confidence:** 3
**Soundness:** 4 excellent
**Presentation:** 3 good
**Contribution:** 4 excellent

**Summary:**

The authors of this paper propose a linear-time algorithm for maximizing monotone and submodular functions under knapsack constraints. The proposed algorithm extends the threshold-greedy algorithm for cardinality constraints. Besides the theoretical results, the authors also provide experimental results to demonstrate the advantages of the proposed algorithm compared to SOTA algorithms.

**Questions:**

Is the DENSITYGREEDY algorithm in experiments also accelerated by the Lazy Greedy idea?

**Limitations:**

1. This paper discusses the situation of minimizing the number of queries to the objective function. In many important applications of submodular optimization, such as influence maximization, we may first generate a sketch data structure that can well approximate the objective function and evaluating the approximate function value is not that computational expensive. It would be better if the authors could discuss something about this in Conclusion.
2. One another interesting point is that the objective submodular function is often hard to compute exactly. What if we only have a noisy oracle of the function value? Is the proposed algorithm robust to a noisy oracle?
3. A minor issue. After Observation 3.3, the authors say "Let l be the value of k". I guess here "k" should be "h" since there is no parameter k in Algorithm 1.

**Strengths And Weaknesses:**

1. The proposed algorithm has linear time complexity and a strong approximation ratio.
2. The authors also provide strong information-theoretic lower bounds for the problem studied in this paper.
3. The authors conduct experiments on real-world datasets to show the practical advantages of the proposed algorithm to SOTA algorithms.

---

> ### Author Response · Authors · 2022-08-02
> **The DensityGreedy algorithm in the experiments is accelerated by the Lazy Greedy idea**
>
> We have used the lazy update idea for the DensityGreedy algorithm. Furthermore, in order to be fair and provide a better comparison, in our experiments we also used a version of the BoostRatio algorithm [36] with lazy updates. Note that the original implantation provided for BoostRatio in [36] does not use lazy updates. In the code attached to the submission, we have provided the implementation of the algorithms with lazy updates (suffixed with *_optimized*). In our experimental results, we have used those implementations.

---

> ### Author Response · Authors · 2022-08-02
> **Discussion regarding the case we have access to a noisy oracle**
>
> Thanks for pointing this out. We would like to mention the following points:
> * In this paper, following most works in the field, we take the number of oracle queries as a measure for the time complexity.
> * In most existing algorithms, the time complexity of the other operations in the algorithm match, up to poly-logarithmic factors, the number of query complexity. Therefore, an algorithm with less query complexity most probably results in an algorithm with less wall-clock time complexity as well.
> * There are a few works such as [a] that considered erroneous oracles, and extending our work in this direction is an interesting question for future research. We would make sure to include this point in the conclusion to the revised version.
>
> [a] Hassidim Avinatan, and Yaron Singer. "Submodular optimization under noise", COLT 2017.

---

> ### Author Response · Authors · 2022-08-02
> **Typos after Observation 3.3**
>
> Thanks for pointing this out. We will fix this issue in the revised version of  the manuscript.

---

### Meta-Review · Area_Chair_6KyE · 2022-08-25

**Recommendation:** Accept
**Confidence:** Less certain

**Metareview:**

Overall, this paper achieves strong and interesting results regarding the query complexity of submodular maximization. One reviewer was concerned that the lower bound result was maybe a folklore result that is easy to prove. However, sufficient evidence to justify that claim was not provided and other reviewers did not share the same concern.

**Award:**

No

---

### Decision · Program_Chairs · 2022-09-14

Accept